# Indoor Environmental Quality Assessment and Occupant Satisfaction: A Post-Occupancy Evaluation of a UAE University Office Building

Young Ki Kim [1,*], Yasmin Abdou [1], Alaa Abdou [2] and Hasim Altan [3]

1 Department of Architectural Engineering, United Arab Emirates University, Al Ain 15551, United Arab Emirates; 201450077@uaeu.ac.ae
2 Department of Architecture, Ajman University, Ajman 346, United Arab Emirates; alaa.ibrahim@ajman.ac.ae
3 Department of Architecture, Arkin University of Creative Arts and Design, Kyrenia 99300, Cyprus; hasimaltan@gmail.com
* Correspondence: kim9519021@gmail.com; Tel.: +971-3-713-5330; Fax: +971-3-713-4990

**Abstract:** As occupants spend almost 90% of their day indoors, especially in the workplaces, Indoor Environmental Quality (IEQ) plays a primary role in health and wellbeing, productivity, and building energy consumption. Adopting the IEQ and Post-Occupancy Evaluation (POE), data has been gathered from nine multilevel open offices within a university building located in Al Ain, in the United Arab Emirates (UAE) for three winter months. Physical parameters were monitored using data loggers to record the main IEQ factors. In parallel, POE questionnaires have been distributed to obtain occupants' satisfaction with the IEQ and health-related symptoms experienced in the workspaces. The IEQ and POE data have shown slightly above or below the recommended ranges with the occupants similarly and slightly dissatisfied with the building. The thermal comfort revealed concerns with 99% of temperatures below international standards where 55% of the survey respondents reported "too cold". The IAQ measurements showed 45% and 30% of the respondents reporting "stuffy air" and "headache" which indicated symptoms that could be tracked to other parameters or a combination of several, and the findings have been discussed in detail in this paper. This research contributed to identifying correlations between measured data and occupant satisfaction and identifying common IEQ defects and their sources to better communicate with facility managers and architects.

**Keywords:** indoor environment quality; post-occupancy evaluation; occupant satisfaction; indoor air quality; university office building; hot arid climate

## 1. Introduction

Through the ongoing study of climatic trends, rising extreme weather events, along with depletion of the ozone layer, scientists are anticipating a large set of negative effects on natural and human systems [1–3]. A number of detected elements are causing a level of concern as their exposure is projected to increase with the escalating climatic conditions. These include overall heat, ultraviolet penetration, an increase of biological materials such as pollen, mould, and infectious agents, and air pollutants, especially particulate matter [4,5].

To avoid these irregularities in the outdoor environment, people tend to spend most of their day—irrespective of office work—indoors. In the United States, people spend approximately 90% of their time in the indoor environment [6,7]. This emphasizes the importance of the indoor environment quality (IEQ) within buildings. Moreover, IEQ has a set of direct and indirect effects not only on the health and wellbeing of people inside the building but also the productivity levels, especially in the office or workplace. Research shows that achieving favourable indoor climate conditions in the workplace can reduce

employee absenteeism, reduce staff turnover, and increase occupants' productivity and satisfaction [8–11].

The IEQ importance has been seen in several building standards ranging from the UAE's regional Estidama mandatory building standards to the LEED green buildings standards [12,13]. Although all new office buildings adhere to the mandatory standards, no efforts have been put into carrying out post-occupancy evaluations to validate these buildings' performance after they have been built. Many factors may cause differences between actual and predicted performance anticipated in the design or modelling process. These include differences in operational practices and schedules, changes that occurred during construction, occupancy patterns and densities, and other issues that cannot be precisely foreseen [14,15]. POE can serve a great role in measuring, analysing, and enhancing the building's performance and occupant satisfaction regarding IEQ [16,17]. Unfortunately, POEs of buildings are not commonly conducted or shared, especially in such a hot and arid desert climate, as the context adds several complications to the occupancy patterns and overall building performance [18].

The goal of this study is to undertake a comprehensive POE of a case study of a higher education office building in the UAE, aiming to measure, analyse, and compare perceived and actual results regarding IEQ and occupant satisfaction. The findings of this study would add an important data point to existing research on POE in hot and arid desert climates, revealing its IEQ trends, causes of occupant dissatisfaction, and prevalence of building-related health symptoms.

### 1.1. IEQ at the Workplace

Since the early 2000s, studies on IEQ display a direct relationship to outdoor environmental conditions [19,20]. Presently the outdoor environment detects a concern regarding the increasing concentration of several harmful pollutants due to global warming and climate change. Higher pollutant concentrations were correspondingly found in the indoor environment; some of which include $NO_x$, $SO_2$, $O_3$, CO, volatile and semi-volatile organic compounds (VOCs), particulate matters (PMs), as well as microorganisms [21–23]. These contaminants can cause a set of health symptoms in humans that vary in severity; according to each's toxicity, concentration, and exposure time [24,25]. A common effect of these contaminants' exposure is known as Sick Building Syndrome (SBS) [26]. SBS is experienced when people show a series of uncomfortable health-related symptoms. These symptoms include eye, nose, and throat irritations, allergies, headaches, fatigue, asthma-like symptoms, and several more [27,28]. Although people may not know the exact reasons for these syndromes, the syndrome may disappear once the affected person leaves the office or building [29].

Several studies at workplaces find that better IEQ decreases the risk of experiencing SBS and increases user comfort which in turn increases individual work productivity [30–32]. The increase in work productivity comes with an increase in economic benefits to companies, universities, or schools. Studies show green buildings to have significantly higher rates of occupant satisfaction when it comes to IEQ as well as allow a reduction in energy consumption [33–35]. Focusing on IEQ can reduce unnecessary energy costs while having a positive effect on thermal comfort which in return can optimize work productivity [36].

Several factors contribute to IEQ including ventilation and it can vary from different ventilation typologies, facility management, and occupant behaviour. How well air conditioners are being maintained and operated affects IAQ and overall IEQ [37]. For example, having a good air filtration system will help to significantly reduce the amount of PMs or fine dust that enters the building from the outside, thus providing better air quality for the occupants. This approach stresses the quality of the ventilation system used, the facility management methods, as well as the indoor occupant's behavioural activities. Moreover, a study that focuses on assessing the IAQ and ventilation rate in schools, finds relations concerning several pollutants exposure and health symptoms experienced [38]. The most-reported health symptoms were found to be SBS and asthma. Investigations

show responsible pollutants to be TVOCs, and allergens that were measured in floor dust. This IAQ assessment demonstrates how low ventilation rates increase health risks among all building occupants.

Another study compares green buildings and non-green buildings through quantitative measurements of IEQ and qualitative occupant satisfaction surveying. The study finds better quantitative performance for the green building in terms of IEQ factors and pollutant concentrations, which translates to better occupant satisfaction and fewer acute health symptoms experienced by occupants in the better IEQ conditions of the green buildings [39].

A commonly seen strategy in several previous studies on IEQ [11,14,38] includes the building's occupant perception as a form of investigative POE study to enhance IEQ. Investigative POE is when a correlation between physical environmental measures and subjective occupants' response measures is studied [40]. POE can be in the form of questionnaires, surveys, or interviews that focus on subjectively measuring certain IEQ criteria or satisfaction levels. Benefits of performing POE include identifying and resolving issues regarding user comfort, overall satisfaction, and productivity [41]; as well as offering documentation as direct input to create a feedback loop for future building cycles [40,42]. Challenges, however, can be linked to the instrument's reliability and the confounding IEQ variables and correlations [43].

This briefly summarizes the findings of the literature review of previous research related to the topics of IEQ and POE. More details on each study and its, date, location, focus, and relative findings can be found in Table 1.

**Table 1.** Literature review findings on IEQ effects at the workplace.

| Title of the Study | Region Studied | Study Focus | Key Findings | Year |
|---|---|---|---|---|
| Occupant productivity and office indoor environment quality. | - | Literature study. | The literature review shows both the economic and health related benefits of good IEQ. It illustrates the significance of the impact of the IEQ on occupant comfort and productivity. | 2016 [8] |
| Satisfaction of occupants toward indoor environment quality of certified green office buildings in Taiwan | Taiwan | A post-occupancy evaluation was employed in the study consisting of a field survey of subjective perception among indoor occupants and on-site environmental measurements. | The overall IEQ satisfaction was statistically significantly greater in the certified green building than the conventional buildings. However, a re-visit of thermal comfort-related criteria may be required. | 2014 [11] |
| Spatial mapping of occupant satisfaction and indoor environment quality in a LEED platinum campus building. | USA | POE approach with GIS-based spatial mapping method was used to analyse and visualize the survey results of building occupant satisfaction and the measured indoor environment quality. | Occupants complained regarding thermal comfort, reporting it was too cold. $CO_2$ level was also predominantly higher. Light levels in the building were found to be higher than preferred as artificial lighting was excessively used even when daylight was available. | 2014 [14] |
| Patients and the sick building syndrome. | USA | Suggest physician approaches to identify disease in individual and group effects on patients and analyse the impact of indoor environmental exposure. | Sick building syndrome can show several recognizable symptoms that include eye irritation, nose irritation, throat irritation, headache, fatigue, asthma-like symptoms, and more. | 1994 [27] |

**Table 1.** *Cont.*

| Title of the Study | Region Studied | Study Focus | Key Findings | Year |
|---|---|---|---|---|
| Indoor Air Quality in the 21st Century: Search for Excellence. | - | Studies key principles for a new philosophy of excellence. | Improved indoor air quality increases productivity and decreases sick building syndrome symptoms. Individual control of the thermal environment should be provided to increase user comfort. | 2000 [30] |
| Comparative study on the indoor environment quality of green office buildings in China with a long-term field measurement and investigation. | China | This study analyses the subjective questionnaires and objective measurements of the indoor environment quality (IEQ) in green building. | Results show that the green buildings in China possess significantly higher IEQ satisfaction levels than conventional buildings. This emphasises the importance of operation management and individual control methods in the building. | 2015 [33] |
| Thermal comfort and behavioural strategies in office buildings located in a hot-arid climate. | Australia | The effects of indoor climate on thermal comfort levels and adaptive behaviour of office workers. | Shows office workers prefer adjusting the set temperature of the building to 22.21 °C for both seasons. As opposed to the ASHRAE scale, it occurred at 20.31 °C in winter. Further research can reduce overcooling cost with a positive effect on thermal comfort and workplace productivity. | 2001 [36] |
| Perception of indoor environment quality in differently ventilated workplaces in tropical monsoon climates. | Sri Lanka | The research investigates the perception of indoor environment quality (IEQ) in differently ventilated workspaces. | Air conditioning (AC) and ductless mini split system air conditioning (MM) buildings were rated more satisfactory than naturally ventilated (NV) systems for overall comfort of indoor environment conditions. | 2015 [37] |
| Indoor air quality, ventilation, and health symptoms in schools: An analysis of existing information | USA | investigates causal relationships between health symptoms and exposures to specific pollutants in schools | Reported ventilation and $CO_2$ data strongly indicate that ventilation is inadequate in many classrooms, possibly leading to health symptoms. | 2003 [38] |
| Indoor environmental quality, occupant satisfaction, and acute building-related health symptoms in Green Mark-certified compared with non-certified office buildings | Singapore | This study compared IEQ performance in green and non-green office buildings. Adopting a cross-sectional study design between objective measurements and subjective measurements. | This study offered a positive association of green buildings with qualitatively and quantitatively measured performance of IEQ. | 2018 [39] |
| Listening to the occupants: a Web-based indoor environmental quality survey | - | Developing a benchmarking survey that can be used as a diagnostic tool to identify specific problems and their sources | The research discusses survey guidelines to create a feedback loop for building industry professionals, so that they can learn how various building design features and technologies affect occupant comfort, satisfaction and productivity. | 2004 [41] |

| Title of the Study | Region Studied | Study Focus | Key Findings | Year |
|---|---|---|---|---|
| Measured energy use and indoor environment quality in green office buildings in China. | China | Energy consumption and indoor environment quality (IEQ) are compared in green office buildings with common ones through energy data collection, physical parameters measurement and satisfaction survey. | User satisfaction in green buildings is statistically significantly higher than those in common buildings. Especially in the field of thermal environment, IAQ, facilities and operating & maintenance. | 2014 [44] |

### 1.2. IEQ Factors

As stated in the Indoor Environment Handbook, IEQ includes four main factors of IAQ such as thermal comfort, lighting quality, and acoustic quality [45]. Each is measured by a set of parameters, and has several control methods and related issues; which are summarized in Table 2.

**Table 2.** IEQ factors, parameters, control methods, issues, threshold, and health.

| IEQ Factors | Parameters | Control Method | Issues | Parameter Measures | Threshold | Health Symptoms |
|---|---|---|---|---|---|---|
| Thermal Comfort | Temperature Relative humidity Air velocity User activity | Air conditioning system Building design | Adaptation Building integration Energy use | Temperature | 24–26 °C ** | Respiratory problems |
| | | | | Relative Humidity | 30–60% ** | Microbial growth, skin drying, irritation of mucus membranes, and dry eyes |
| Indoor Air Quality | Pollution sources Ventilation rate and efficiency | Source control Ventilation system maintenance | Pollution Fine dust | PM2.5 | 15 $\mu g/m^3$ * | Respiratory and cardiovascular diseases including asthma, myocardial ischemia, high blood pressure and heart disease |
| | | | | PM10 | 50 $\mu g/m$ * | |
| | | | | $CO_2$ | 800 ppm * | Increased risk of sick building syndrome and symptoms such as headache |
| | | | | TVOCs | 312 ppb * | Dry throat, runny nose, asthma attacks, poisoning, and cancer |
| Lighting Quality | Luminance Reflectance Colour, temperature View, and daylight | Luminance distribution Artificial lighting and daylighting integration | Daylight relation to thermal comfort Energy use | Lux level | 300–500 lux * | Headaches, circadian phase disruptions, breast cancer, sleep disorder, and depression |
| Acoustical Quality | Sound level Absorption Sound insulation Reverberation time | Acoustical control Passive noise control Active noise control | Vibrations and annoyance long term health effects | Sound level | 55 dBA * | Hypertension, stress, poor concentration, memory retention and mental arithmetic |

\* WELL Standard V 2.0. ** ASHRAE Standard 55.

Thermal comfort is a subjective evaluation of one's satisfaction with the thermal environment [46]. It can differ from the perception of one person to another according to a

set of factors such as age, gender, activity level, clothes, etc. Generally, thermal comfort is the most responsible factor for human health, well-being, and productivity. As it has a direct effect on the body's respiratory system. For example, too cold an environment or highly fluctuating temperature can trigger asthma and flu symptoms.

IAQ is an essential factor to assess the quality of the air within a building. health and well-being. The building's ventilation system may be the most underestimated aspect of the indoor air pollution level. Thus, the design and maintenance of such systems are vital.

Sustaining comfortable lighting levels is another crucial factor in the work environment. Major lighting issues need to be avoided such as excessive lighting, glare, flickering, reflection, inconsistent distribution, and lack of integration of daylighting and artificial lighting. Focusing on these issues creates a comfortable workplace environment that increases work productivity. Moreover, user control further increases the lighting quality of the indoor space.

To make an indoor space perform better acoustically, control strategies can be implemented to limit unwanted noise and reverberation. Simple strategies involve using absorbing material, closing sound leaks, reducing contact sound transmission, and/or applying active noise control. Long-time exposure to disturbing noises can lead to a range of health issues such as stress, poor concentration, and productivity losses in the workplace.

### 1.3. IEQ Measurement Parameters and Thresholds

As explained previously, each IEQ factor can be measured by certain parameters. Thermal comfort can be measured by the indoor temperature in degrees Celsius and relative humidity percentage (RH%). IAQ can be assessed by the amounts of fine and coarse particulate matter (PM2.5, PM10) measured in $\mu g/m^3$, carbon dioxide ($CO_2$) measured in ppm, and total volatile organic compounds (TVOCs) measured in ppb. Lighting can be measured by Lux, and acoustics provide the maximum allowed values for each parameter for office buildings [14]. Threshold values along with health effects can be found in Table 2.

## 2. Methods and Materials

Objective and subjective assessment methods were implemented in parallel from December 2019 to February 2020. Continuous and intermittent field measurements have been arranged through this period to measure physical parameters in accordance with the four IEQ factors mentioned earlier; thermal comfort, IAQ, lighting quality, and acoustical quality. Subjective questionnaires were similarly developed to obtain data relative to the occupant's satisfaction with the IEQ factors, and health-related symptoms experienced in the workspaces. A higher education office building has been selected for this study located in Al Ain, UAE, which is characterized by a hot and arid desert climate.

### 2.1. Case Study Building

The case study building selected in the United Arab Emirates University Campus, Al-Ain, UAE. Al-Ain climate is characterized to be a hot and arid desert climate characterised by its long, extremely hot summers (38 °C average) and warm winters (18 °C average), with average relative humidity of 60% [15]. The annual temperature and solar radiation data are illustrated in Figure 1. It shows hot summer and mild winter temperatures (Figure 1a), and high solar radiation reaches regardless of the season in Al Ain, UAE (Figure 1b).

It is a higher education office building named the F1 building, and it houses three colleges (College of Engineering, College of Science, and College of Food and Agriculture) through its three floors with an estimated number of 600 occupants that include faculty, students, researchers, and other staff (Figure 2).

The air conditioning (AC) system consists of 13 air handling units (AHU) located on the building's roof and controlled using a variable air volume (VAV) system. More technical information about the building can be found in Table 3.

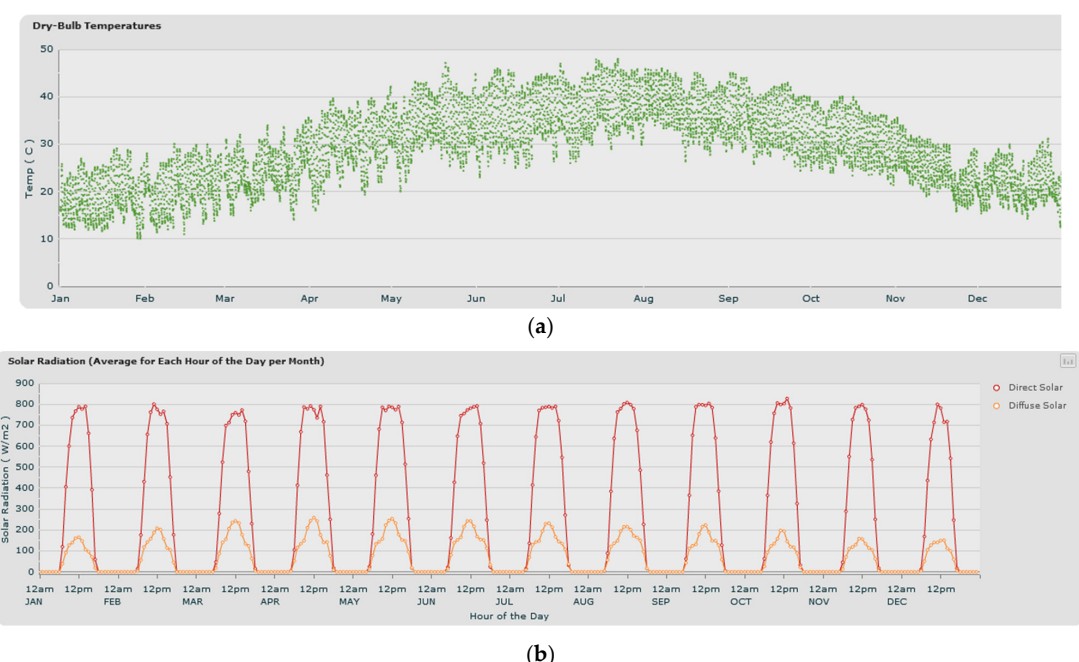

**Figure 1.** Annual weather data for Al Ain, UAE. (**a**) Annual hourly temperature data for Al Ain, UAE (source: Al Ain Airport); (**b**) Solar Radiation data (source: Al Ain Airport).

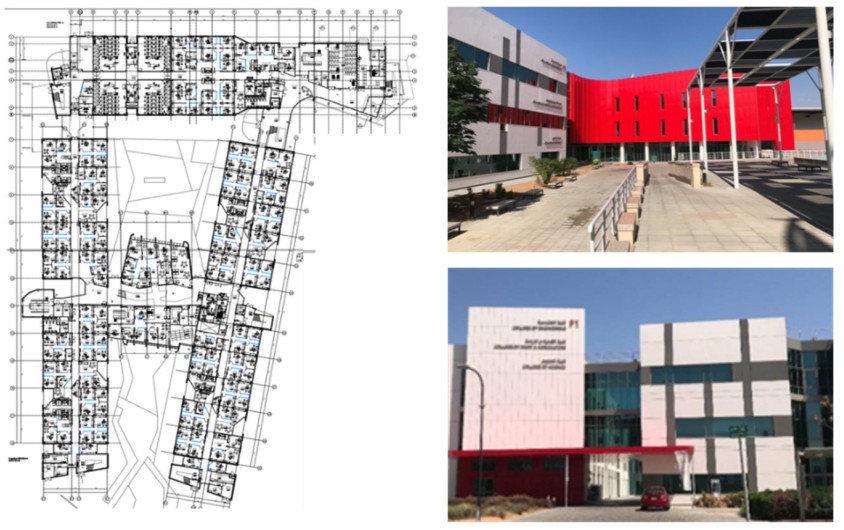

**Figure 2.** Case study building, floor plan, and photos.

**Table 3.** Case study building information.

|              |                   |                                |
| ------------ | ----------------- | ------------------------------ |
|              | Site              | P.O. BOX 15551, Al-Ain, UAE    |
|              | Use               | Office, labs, and lecture rooms |
| Architecture | Building area     | 7120 m²                        |
|              | Gross floor area  | 21,360 m²                      |
|              | Cooling           | Campus district cooling        |
| Mechanical   | Air Handling Unit | 13 AHUs on the roof            |
|              | Control           | VAV                            |
| Electrical   | Lighting          | T5 flounce lamp (office)       |
|              | Control           | Wall switches on/off           |

### 2.2. IEQ Monitoring

To assess the IEQ of a building objectively, continuous measurements were performed to obtain profiles of air temperature (°C) and relative humidity (RH in %), using the HOBO U12 data logger in selected zones at fifteen-minute intervals for the whole period from December 2019 to February 2020. Spot measurements were also performed to identify concentrations of particulate matter (PM2.5 and PM10 in $\mu g/m^3$), carbon dioxide ($CO_2$ in ppm), and total organic volatile compounds (TVOCs in ppm) in the indoor air. These IAQ parameters were collected in a walkthrough manner at fifteen-minute intervals, randomly selected 5 working days in each month, during the working hours from 8:00 am to 4:00 pm, using the air mentor pro (Model No: 8096-AP). For the level of lighting and noise exposure in the workspace, a hand-held environment meter was used to obtain a realistic approximation of the illumination (in lux) and noise levels (in dB) received by the employee using the PRECISION GOLD Multifunction Environment Meter (Model N09AQ) via a walkthrough manner similar to like the air mentor pro. Table 4 shows more details about the devices used in this research.

**Table 4.** Devices used, parameters measured, and measuring intervals.

| IEQ Factor | Device | Image | Parameters Measured | Range | Accuracy | Measuring Intervals |
|---|---|---|---|---|---|---|
| Thermal Comfort | HOBO | | Temperature Relative humidity | Temperature: −20 °C to 70 °C RH: 5% to 95% | Temperature: ±0.35 °C RH: ±2.5% | 15 min |
| IAQ | Air Mentor Pro | | PM2.5 PM10 $CO_2$ TVOCs | PM2.5 Range: 0~300 $\mu g/m^3$ PM10 Range: 0~300 $\mu g/m^3$ $CO_2$ Range: 400~2000 PPM TVOC Range: 125~3500 PPB | N/A | 15 min |
| Lighting Quality | PRECISION GOLD Environment Meter | | Lux level | Lux: 0 to 20,000 Lux | Lux: ±5% rdg +10 dgts | 15 min |
| Acoustical Quality | | | Sound level | Sound: 35 to 130 dB | Sound: ±3.5 dB | 15 min |

To sufficiently analyse and compare the data, a total of 12 HOBO devices, 6 air mentor devices, along with 1 handheld environment meter device have been set up throughout the three floors of the F1 building. Open workspace zones were selected for analysis appropriately. Figure 3 shows the placement of each device on the floor plan.

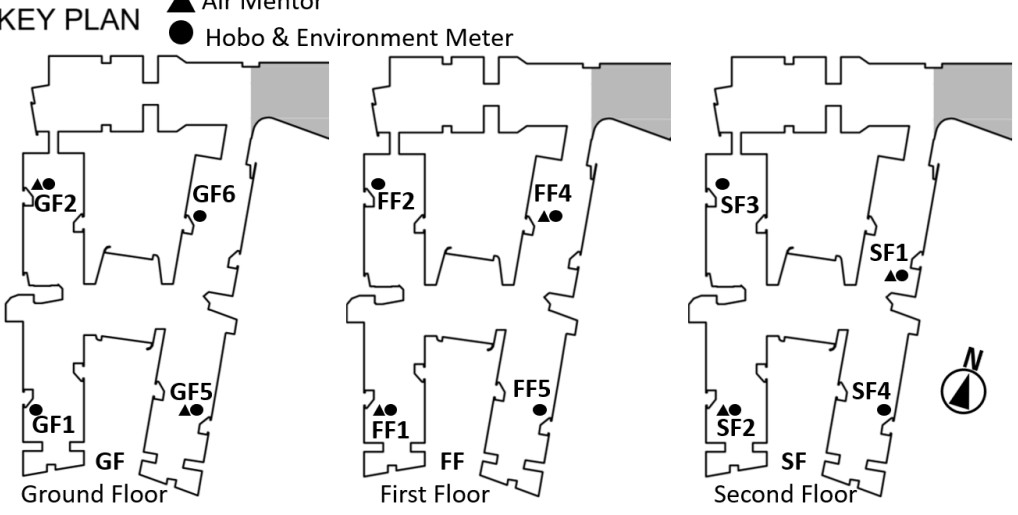

**Figure 3.** Monitoring device locations in the floor plan.

On each floor, 4 zones were selected for continuous measurements, 2 of which were selected for spot measurements as a 5-day monitoring spot. The selected zones were based upon a characterization set: to have similar furniture layout, lighting arrangement, exposure to noise, and the same level of control over temperature, lighting, and ventilation. In each selected zone, the sampling location of the measuring instrument was specified at the height of the breathing zone of the seated employees and away from direct sunlight which may affect the accuracy of the readings. One additional sampling point for outdoor air was also included to represent outdoor air quality at the time of indoor spot measurements. Furthermore, interviews were conducted with the facility management team to obtain building operation schedules and technical details.

### 2.3. User Satisfaction Questionnaire

As part of this investigative POE study, a survey questionnaire was developed with the aim of exploring the perception of the building's occupants and detecting sources causing discomfort and dissatisfaction with the office IEQ. To develop the questionnaire, face-to-face meetings and e-mails with the facility management team have taken place to better understand how the building operates and manages. A paper-based questionnaire was developed accordingly and followed the same four measured IEQ factors. The questionnaire was then reviewed by three academic researchers experienced in questionnaire design. Their feedback was considered for improving the questionnaire content.

During the actual survey, the questionnaire was delivered by the surveyor in person to the building occupants who volunteered to participate anonymously to ensure personal data protection. The surveyor provided a brief introduction and explanation of the aim of the survey along with any clarifications if needed. Respondents were asked how often they experience a set of sources contributing to uncomfortable sensations. As mentioned earlier, the questions followed a standardized format consisting of a 4-point scale from which the respondent can choose a rank equivalent to their experienced sensation. The options provided were: never (neutral), sometimes (slightly uncomfortable), regularly (moderately uncomfortable), and often (extremely uncomfortable). For analysis, the Likert scale was applied, and the rank selected from the 4-point scale in each question was transformed into an integer between 0 and 3.

The survey questions were organized into 3 main sections. Starting with a background section that includes occupants' demographics and job nature. Secondly, the occupant's perception of the sources contributing to un-comfort and dissatisfaction with the indoor environment quality, and their frequency of occurrence were investigated. The final section included questions related to the prevalence of health-related symptoms experienced in the workplace that are directly linked to the overall IEQ. The objective of the questionnaire arrangement was to facilitate a reasonable connection between quantitative and qualitative data. For further evaluation of location influences, there was a corresponding question about the respondent's office location. Lastly, an open-ended question was provided to gain additional comments or suggestions about IEQ. The POE survey questionnaire can be found in Appendix A.

### 3. Results and Analysis

#### 3.1. IEQ Monitoring Data

In this study, environmental monitoring was performed across selected open workspace zones to act as a physical reference to the sources that impact the IEQ and occupant satisfaction. The following discussions focused on describing the overall statistical trends in the distribution of the measured parameters and compared them primarily to the international standards.

#### 3.1.1. Thermal Comfort

Thermal comfort was monitored by recording temperature and relative humidity (RH) values continuously. 1-day profiles of air temperature and RH during a randomly selected

working day (5 January) were derived from the measurements, depicted in Figure 4, across 24 h from a total of 12 studied open office zones throughout the office building. On that day, the temperature ranged from almost 20 °C to 23 °C with an average of 21.3 °C. There is a natural increase in temperature during the highlighted working hours as the building is mostly occupied, however, all zones appear to be far below the recommended range from 24 °C to 26 °C as suggested by ASHRAE 55. RH percentage ranged from 48% to 62% with an average of 55.8%, most of the measured zones were revealed to be within the standard recommendation of 30% to 60% as suggested by ASHRAE 55.

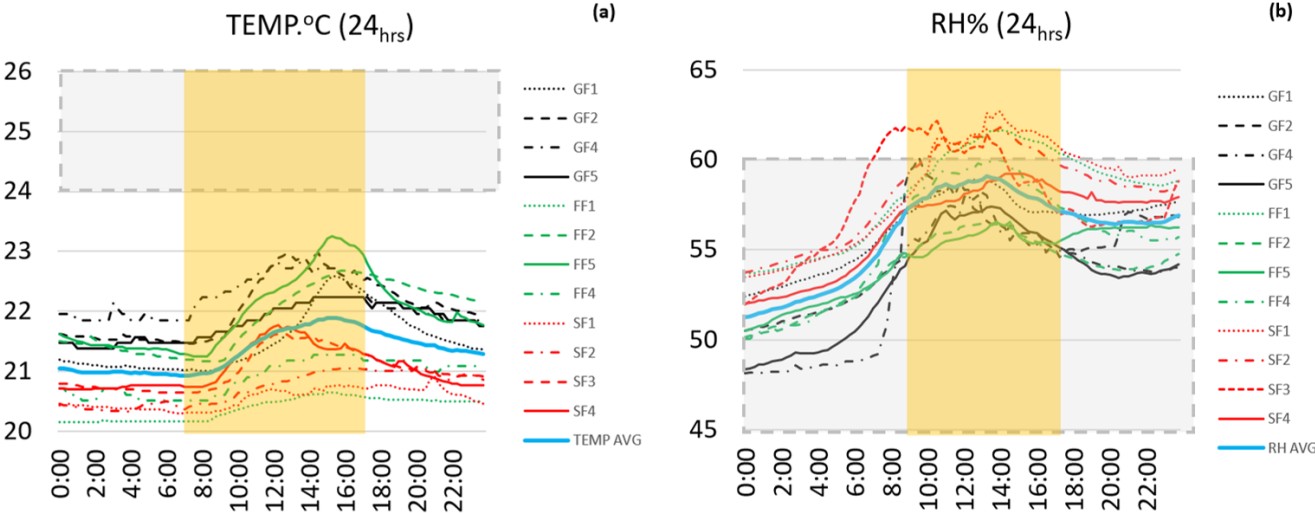

**Figure 4.** Temperature (**a**) and RH (**b**) 1-day (24 h) profiles. The blue line indicates total readings 'average, the yellow box denotes working hours, and the grey box indicates recommended thresholds.

To put the 1-day profile into a larger perspective, full period (December–February) temperature and relative humidity monitoring data have been plotted in Figure 5. Through the 3 winter months, the temperature ranged from a minimum of 19.4 °C and a maximum of 24.3 °C with an average of 21.3 °C (December), 21.1 °C (January), 21.9 °C (February), and 21.5 °C total average. All of which are below the standard threshold highlighted in grey (Figure 5a). These values indicate a cause of discomfort and possible occupant dissatisfaction regarding the indoor temperature. On the other hand, RH% ranged from a minimum of 29% and a maximum of 77% with averages of 57.3% (December), 53.2% (January), 48.5% (February), and 53% total average. Although there were a few extremities, most values were revealed to be within the threshold highlighted in grey (Figure 5b).

### 3.1.2. Indoor Air Quality

The IAQ assessment was covered by recording spot measurements of the concentrations of PM2.5, PM10, $CO_2$, and TVOCs in the air. There were no major flaws exhibited by the results; however, design factors and occupant behavioural trends have been noted. The box plots in Figure 6a–d represent each variable's recorded measurements across 6 studied open office zones and their total average (shown in dark grey). Starting off with PM2.5 (Figure 6a), the results appear to be lower than 15 μg/m³ as suggested by the WELL building standard, with a total average of 4.34 μg/m³. Notably, however, zone SF1 (located on the second floor) recorded the largest variance in concentrations as well as the highest recorded PM2.5 concentration of 26 μg/m³ with a slightly higher average of 5.2 μg/m³. This can be directly linked to this open office's spatial design which is larger in area and has a higher rate of occupancy compared to the rest of the open offices examined.

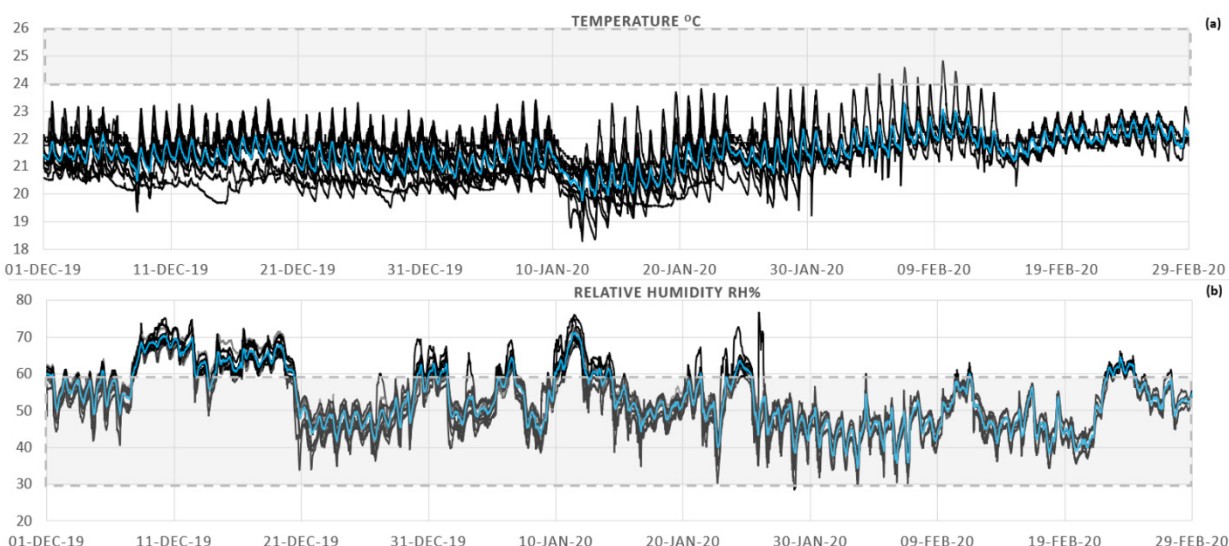

**Figure 5.** Temperature (**a**) and RH (**b**) December to February profiles. The blue line indicates average, and the grey box indicates recommended thresholds.

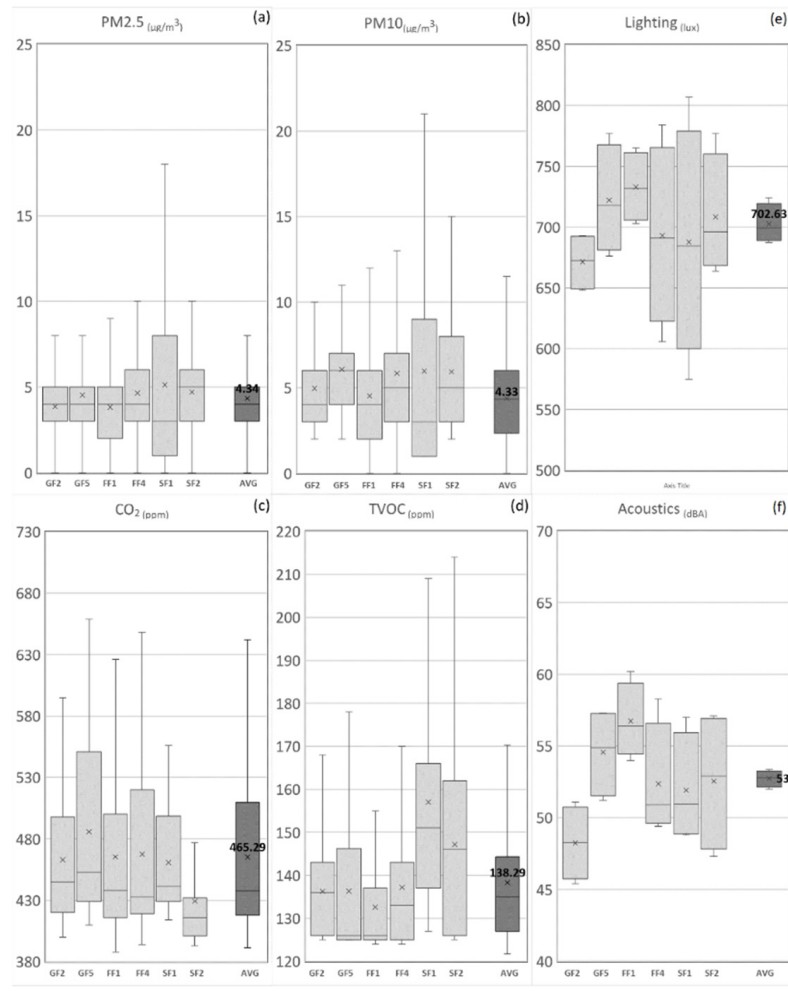

**Figure 6.** Distribution of environmental parameters obtained from spot measurements during December to February (dark grey box indicates total readings' average) (numbers indicated are mean values).

Similarly, PM10 (Figure 6b) results recorded lower than 50 μg/m$^3$ as suggested by the WELL building standard, with a total average of 4.38 μg/m$^3$. Again, zone SF1 showed great variance and recorded the highest of 33 μg/m$^3$. This highlights the relationship between the open office design layout and the particulate matter concentration levels in the air.

As for the $CO_2$ concentration (Figure 6c), the total mean concentration was found to be 465.3 ppm, which is lower than the 800 ppm suggested by the WELL building standard. Notably, zone SF2 recorded the lowest and most consistent $CO_2$ concentration ranging from 494 ppm to 477 ppm due to the high level of mechanical ventilation rate.

For TVOCs concentration (Figure 6d), the total average result was revealed to be 138.3 ppm, which is again lower than 312 ppm as suggested by the WELL building standard. This was expected as the building is relatively old and has no new materials introduced that may increase TVOCs' concentration. Concurrently, both zones located on the second floor (SF1 and SF2) show higher variance in measured concentrations of TVOC with the highest recorded values of 209 ppm and 219 ppm, respectively. Overall, all IAQ parameters (Figure 6a–d) recorded were lower than the thresholds suggested by the WELL building standard.

### 3.1.3. Lighting Quality

Lighting levels have been recorded across 6 open offices as spot measurements throughout the office building. Figure 6e depicts box plots of each individual zone studied as well as the total average box plot (shown in dark grey). The total average is 702.63 lux, which is higher than 300–500 lux, as recommended by the WELL building standard. Measurements vary broadly between each open office as they are greatly affected by different orientations since each open office integrates large windows. Moreover, none of the open offices achieve the recommended range of lux level, this is particularly due to the imbalance between artificial lighting (T5 flounce lamp) and daylighting (windows) which are usually both used simultaneously in an excessive manner causing relatively uncomfortable lighting. Suggestions to solve such an issue include providing a higher level of control on the artificial lighting system to be able to dim the lights or choose certain fixtures to turn on/off. Additional buildings' annual energy savings, and energy cost savings can be made if this suggestion may be applied, which will all be in favour of the occupants' comfort.

### 3.1.4. Acoustical Quality

Noise levels demonstrate a wide range of variance as it is greatly affected by human factors as well as several machine factors. On average, the level of noise revealed in all open office zones is around 53 dBA (Figure 6f), which is just below 55 dbas as suggested by the WELL building standard. The largest variance of noise level recorded appears to be in the upper floor (zone SF2 ranging from 47.3–57.1 dBA with an average of 52.6 dBA), as it has been observed to have additional noises coming from the HVAC system. Notably, zone FF1 shows the highest range of noise recordings ranging from 54.0–60.2 dBA with an average of 56.8 dBA; this adds a concern regarding the effect of the open office layout and the lack of acoustic absorption materials.

### 3.2. Occupant Satisfaction Questionnaire Data

Although the physical measurements discussed in the previous section show a level of compliance with common environmental standards of office buildings, the results of such measurements do not indicate if the occupants are comfortable with the overall IEQ in these buildings. To develop a comprehensive perspective, a questionnaire survey was implemented and analysed.

An EXCEL spreadsheet was used to analyse and visualize statistically the collected data. A total of 90 occupants' data was gathered and analysed. Table 5 illustrates the demographic information of respondents. The number of females was higher than males, with 58% and 42%, respectively. The age group (20–30 years) formulated the majority of

respondents with 78% of the sample, with research-based job categories composed of 60%. This is mainly because this study focuses on the open office and reception areas, which are mostly occupied by young researchers and fewer administrative or higher academic faculty members.

**Table 5.** POE occupants' demographics.

| Total (N) | Gender | | Age Group | | | Job Description | | | |
|---|---|---|---|---|---|---|---|---|---|
| | **M** | **F** | **20–30** | **30–40** | **40–50** | **Researcher** | **Postgrad Student** | **Specialist Engineer** | **Secretary** |
| 90 | 38 (42%) | 52 (58%) | 70 (78%) | 19 (21%) | 1 (1%) | 54 (60%) | 14 (16%) | 18 (20%) | 4 (4%) |

The data gathered was analysed into 3 categories. The first category illustrates the distribution of the respondents' votes corresponding to the different factors of IEQ. Secondly, a sensitivity analysis was developed for the overall IEQ parameters contributing to the discomfort and dissatisfaction of occupants. Finally, the prevalence of health-related symptoms experienced in the workplace was presented. The above analysis is illustrated in Figures 7–10, respectively.

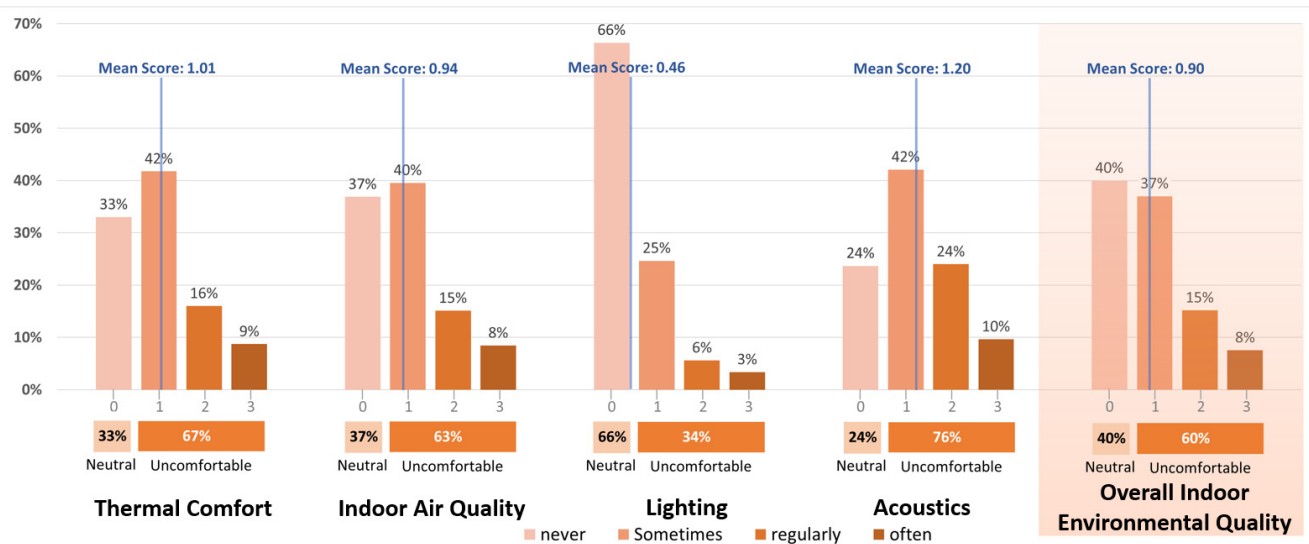

**Figure 7.** Distribution of votes corresponding to the different factors of IEQ as expressed by building occupants in questionnaire surveys.

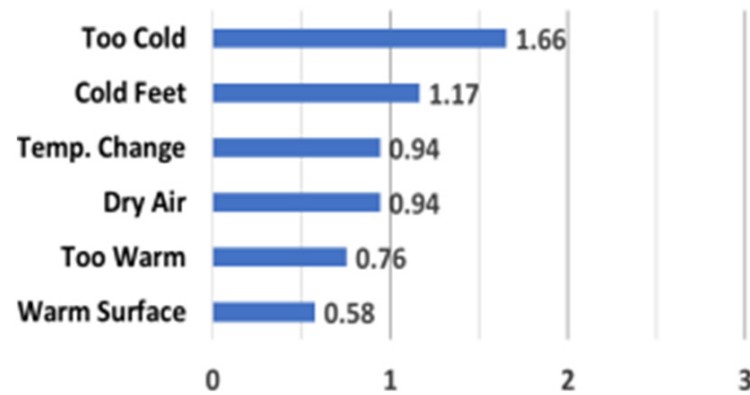

**Figure 8.** Frequency scores of sources contributing to discomfort and dissatisfaction toward thermal comfort.

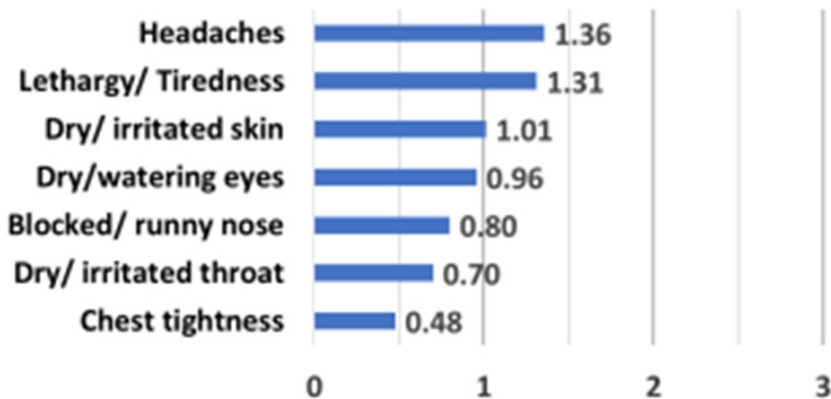

**Figure 9.** Sensitivity analysis of most common health-related symptoms caused by the Indoor Environment Quality.

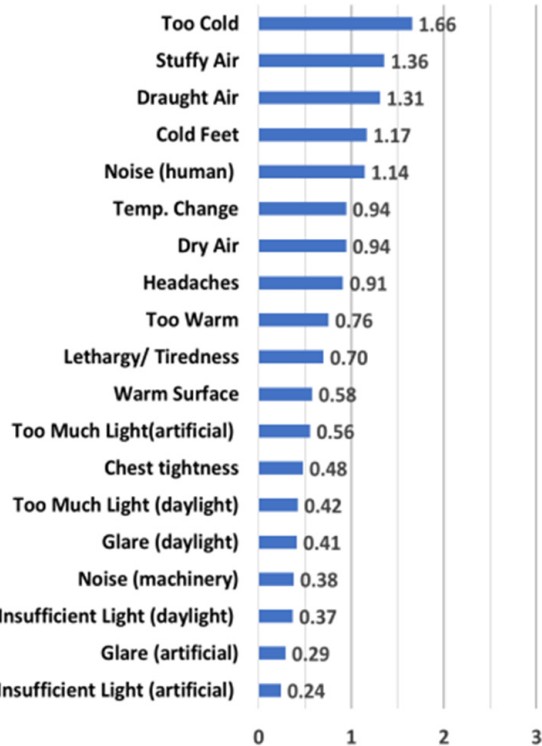

**Figure 10.** Frequency scores of sources contribute to discomfort and dissatisfaction with IEQ.

The questions related to the first category of analysis were interpreted according to the corresponding rank of the vote. The "never" (0) votes were assigned as a single category of "neutral" while votes for "sometimes" (1), "regularly" (2), and "often" (3) were pooled into a single category of "uncomfortable". Figure 7. illustrates the distribution of the respondents' votes regarding the frequency analysis of "neutral" and "uncomfortable" sensations relative to the different 4 factors. The overall respondents' vote concerning the IEQ was presented in the same figure.

The overall analysis reveals that 60% of respondents were uncomfortable with the IEQ in the workplace (Figure 7e), suggesting that in general there are a number of existing issues affecting the comfort of the occupants in the office building. These percentages were further dissected and detailed according to the 4 factors of IEQ in Figure 7a–d and detailed in the forthcoming sections.

### 3.2.1. Thermal Comfort

The analysis of the thermal comfort-related data reveals that 67% of respondents' votes are leaning toward a level of uncomfortable perception, while 33% are neutral (Figure 7a). Moreover, the correlated mean score (1.01) denotes "slightly uncomfortable".

Questions related to thermal comfort focused on the temperature and relative humidity parameters. The sensitivity analysis of sources contributing to discomfort and dissatisfaction is illustrated in Figure 8, in the form of frequency scores. The frequency score was calculated giving a value of never (0), sometimes (1), regularly (2), and often (3). Results were then arranged in descending order from the highest to the lowest score.

Among the parameters of thermal comfort, the analysis revealed that "too cold" scored (1.66) as the highest source of uncomfortable sensation experienced by the respondents. The second highest source was "cold feet" (1.17) similarly emphasising the cold sensation. Although these scores only indicate "slightly uncomfortable", 42% of the received comments from respondents, expressed complaints about feeling too cold in the workplace, interfering with their work productivity, health, and overall comfort. This also matches with the findings of previous research [7], which reveals "too cold" as the highest reported complaint regarding thermal comfort. On the other hand, "warm surface" scored (0.58) as the lowest source among the 6 investigated sources contributing to discomfort and dissatisfaction with thermal comfort (Figure 8).

### 3.2.2. Indoor Air Quality

Concerning indoor air quality, results reveal that 63% of respondents' votes indicated feeling uncomfortable, while 37% are neutral (Figure 7b), with a correlated mean score (0.95) landing close to "slightly uncomfortable" (1). Questions related to the ventilation parameter investigated the occupant's feeling about "stuffy air" and "draught air". Both sources of discomfort and dissatisfaction scored similar results with 1.36 and 1.31, respectively. Ranking second and third most frequent sources (after "too cold") when compared with all the sources contributing to discomfort and dissatisfaction with IEQ, details are illustrated in Figure 9. This result is supported by 21% of the received comments from respondents complaining about inadequate ventilation due to having no operable windows, emphasising the poor ventilation rate of the HVAC system and the need for user control over building outlets.

### 3.2.3. Lighting Quality

Analysis of occupant's perceptions related to the lighting quality demonstrates that only 34% of respondents are uncomfortable, while 66% are neutral (Figure 7c), with a mean score of 0.46 (between "neutral" (0) and "slightly uncomfortable" (1)). The questions related to the sources causing discomfort with the lighting quality were divided among artificial lighting and daylight, assessing the frequency score of too much light, insufficient light, and reflection or glare. Although all responses scored relatively low compared to other sources contributing to discomfort with the indoor environment quality, "too much artificial light" scored (0.56) the highest source related to the lighting quality, and ranked 12 among all other sources (Figure 10). This result is backed up by some of the received comments stating being uncomfortable with the reflection and glare of daylight entering from existing large windows, and the lack of control over it. These were further tracked to respondents situated near southwest/west windows.

### 3.2.4. Acoustical Quality

The analysis of occupant's perceptions related to the acoustical quality of investigated workplace reveals that 76% of respondents are uncomfortable, while 24% are neutral (Figure 7d), with a mean score of 1.19 (between "slightly uncomfortable" (1) and "moderately uncomfortable" (2)) which is the highest among the 4 factors. The questions regarding the acoustical quality investigated the level of noise and its cause, whether human or machinery. Noise from human sources scored significantly higher (1.14) than noise caused

by machinery (0.38), which is expected in the open offices' layout. When compared to all the sources contributing to discomfort with the indoor environment quality, "Noise (human)" ranked 5th among other sources (Figure 10) which indicates an uncomfortable concern by the respondents.

### 3.2.5. Health-Related Symptoms

The last category of survey questions assessed the prevalence of commonly reported health symptoms triggered by different factors of the indoor environment quality, which were selected based on the conducted literature review [7,15,17]. The sensitivity analysis, illustrated in Figure 9, uncovers that the highest two reported health symptoms are "headaches "and "lethargy and tiredness", scoring 1.36 and 1.31, respectively. According to a literature review, these two health symptoms are associated with inadequate ventilation and higher light levels which were reported earlier by respondents. The following two health symptoms that were uncovered by the sensitivity analysis are "dry/irritated skin" and "dry/watering eyes" scoring 1.01 and 0.96, respectively. According to the conducted literature review, these two health symptoms are associated with "dry air", which scored similarly (0.94) as slightly uncomfortable. Figure 9 illustrated the sensitivity analysis of the seven examined health-related symptoms caused by inadequate indoor environment quality.

To recap the analysis of occupants' satisfaction, Figure 10 illustrates the calculated frequency scores of all 19 examined sources that contribute to discomfort and dissatisfaction with indoor environment quality. In addition, the calculated overall mean scores of the four IEQ factors were ranked and shown in Table 6, where the range 0–3 was interpreted as (0) neutral, (1) slightly uncomfortable, (2) moderately uncomfortable, (3) extremely uncomfortable. The highest factor the respondents were uncomfortable and dissatisfied with was the acoustical quality with a mean score (1.2) landing between "slightly uncomfortable" and "moderately uncomfortable". This was expected due to the open offices' layout. This was followed by thermal comfort with a mean score (1.01) denoting "slightly uncomfortable". That was highlighted earlier by "too cold" as the top identified source contributing to discomfort and dissatisfaction. The third-ranked factor was IAQ with a mean score (0.94), very close to "slightly uncomfortable", triggered by the source "stuffy air". Lastly, the lighting quality factor scored the lowest (0.46), almost midway between "neutral" and "slightly uncomfortable". This represents the least complaints received. The overall IEQ received a mean score of 0.9 indicating respondents are slightly uncomfortable/dissatisfied with the quality of the indoor environment.

**Table 6.** Occupants' satisfaction mean score and standard deviation of factors and overall IEQ.

| IEQ Factors | Rank | Mean Score | Std. Deviation |
|---|---|---|---|
| Acoustical Quality | 1 | 1.20 | 0.92 |
| Thermal comfort | 2 | 1.01 | 0.92 |
| Indoor Air Quality | 3 | 0.94 | 0.91 |
| Lighting Quality | 4 | 0.46 | 0.74 |
| Overall IEQ | - | 0.90 | 0.92 |

## 4. Discussion

To analyse the data gathered from both monitoring and surveying comprehensively. The forthcoming section provides a discussion of the relationship between the obtained physical measurements and the analysis of the questionnaire findings. Table 7 combines all the monitoring and survey results alongside and finds the main causes for each IEQ factor coherently. Furthermore, possible suggestions, perceived limitations, and future works are finally discussed.

**Table 7.** Comparative summary of IEQ monitoring and surveying results and main causes (highlighted in bold are the highest numbers signifying concerns).

| IEQ Factors | Parameter Measures | Threshold | % Measured above Threshold | Total Average | Discomfort Sources and Health Related Symptoms | % of Participants' Reports | Overall Mean Score | Main Causes |
|---|---|---|---|---|---|---|---|---|
| Thermal Comfort | Temperature | 24–26 °C | **99%** | **21.5 °C** | too Cold | **55%** | **1.10** | Facility Management and User Lack of Control |
| | | | | | Too Warm | 19% | | |
| | | | | | Temp. Change | 31% | | |
| | | | | | Cold feet | **39%** | | |
| | | | | | Warm Surface | 25% | | |
| | | | | | Runny Nose | 31% | | |
| | RH% | 30–60% | 27% | 53% | Dry Air | 31% | | |
| | | | | | Thermal Comfort | 23% | | |
| | | | | | Dry/Watering eyes | 32% | | |
| | | | | | Dry Skin | 34% | | |
| IAQ | PM2.5 | 15 µg/m$^3$ | 1% | 4.34 µg/m$^3$ | Chest tightness | 16% | 0.94 | HVAC Layout, and User Lack of Control |
| | PM10 | 50 µg/m$^3$ | 0.1% | 4.38 µg/m$^3$ | Stuffy Air | **45%** | | |
| | $CO_2$ | 800 ppm | 0% | 465.29 ppm | Headache | 30% | | |
| | | | | | Lethargy/Tiredness | 23% | | |
| | TVOCs | 312 ppb | 0% | 138.29 ppb | Dry/Watering eyes | 32% | | |
| Lighting Quality | Lux level | 300–500 lux | **100%** | 702.63 lux | Too much light (daylight) | 14% | 0.46 | User Lack of Control and Layout |
| | | | | | Insufficient light (daylight) | 12% | | |
| | | | | | Too much light (artificial) | 19% | | |
| | | | | | Insufficient light (artificial) | 8% | | |
| | | | | | Reflection or glare (artificial) | 10% | | |
| | | | | | Reflection or glare (daylight) | 14% | | |
| Acoustical Quality | Sound level | 55 dBA | 38% | 53 dBA | Noise (human) | 38% | **1.20** | Facility Management and Layout |
| | | | | | Noise (machinery) | 13% | | |

### 4.1. Thermal Comfort

The thermal discomfort was evidently agreed upon with 99% and 27% of the measured temperature and relative humidity data, respectively, below the standard recommended range (as shown in Table 7). In addition, 55% and 39% of the survey participants reported complaints of "too cold" and "cold feet", respectively. These results highlight the correlation between the average temperature value measured (21.5 °C) and the occupants perceived sensation ("too cold"), as well as validate the recommended range standard set by ASHRAE-55 for the winter in hot arid climate (24–26 °C). The overall mean score for thermal comfort satisfaction was 1.10, which falls between "slightly dissatisfied" and

"moderately dissatisfied" according to the 4-point scale survey conducted. Such failure in meeting the occupant's thermal comfort needs can be traced primarily to the facility management as they are responsible for setting the indoor temperature. Moreover, 48% of the survey participants reported not having enough control over the temperature in their workspace, which is another important, often overlooked, factor contributing to their dissatisfaction with the thermal conditions. Bordass and Leaman (1997) [47] defined perceived control as what users can do to adjust their environment if they are not happy with it. Simply providing the buildings' occupants a minor level of control over the indoor temperature can improve their thermal comfort and overall satisfaction with the IEQ; additionally, occupants' productivity, health, and wellbeing still have the potential to be enhanced in this office building. Interestingly, this issue of setting the temperature lower than the comfort range during the winter months appears to be recurrent in several office buildings in different regions found in similar research [14,48].

### 4.2. Indoor Air Quality

The indoor air quality demonstrated very minor issues, with the highest percentage of measurements above the threshold of only 1% (PM2.5) (Table 7). This indicates that the HVAC filtration system is adequately performing and being maintained. Nevertheless, when compared to commonly reported health-related symptoms as associated with responsible factors based on the literature review, noticeably high complaints have been reported, 45% of the respondents reported "stuffy air". Although "stuffy air" here is correlated with the PM concentration, it may be further affected by the ventilation rate and in/out diffusers' layout. This finding was further highlighted as 46% of the survey respondents reported not having enough control over the ventilation of their workspace. Additionally, it has been examined through the initial walkthrough in the building that the windows were mostly fixed with no possibility to allow the occupant to access fresh air except from the entrances or the HVAC set inlets. Allowing the user to have a level of control over the windows, in this case, will help enhance IAQ, overall IEQ, occupants' productivity, and health and wellbeing. Furthermore, the careful layout of the HVAC systems' inlet and outlet could reduce this issue by well circulating the fresh air and removing the stuffy air.

### 4.3. Lighting Quality

Comparing the lighting quality in terms of measured lux levels and perceived occupants' sensations triggered some controversy. Although 100% of the measured lux levels in the open offices were above the recommended range with a total average of around 703 lux (Table 7), survey respondents did not record any major complaints or sources of discomfort from the lighting system. This indicates that the occupants generally prefer higher lighting levels. As window blinds are available, only 18% of the respondents have reported not having enough control over the lighting in their workspace. However, few occupants suggested adding dimming controls and controls over specific fixtures as the current level of control only allows for an on/off switch for all the lighting fixtures in a particular office space.

### 4.4. Acoustical Quality

The acoustical quality assessment demonstrates that 38% of the sound level in office spaces measured above the standard threshold of 55 dBA with a reasonable total average of 53 dBA. On the other hand, 38% and 13% of respondents complained of uncomfortable noise caused by human factors and machine factors, respectively, with a noticeably high overall mean score for acoustical quality satisfaction of 1.20, which falls between "slightly dissatisfied" and "moderately dissatisfied". This interpretation from the occupants can be traced to the open office layout of the studied workspace as they were also situated directly open onto main circulation hallways, allowing for noises to be easily carried through. Adding buffers and high sound insulation materials to the interior workspace design can

be considered to mitigate these noise issues triggering discomfort, headaches, and lack of concentration.

*4.5. Limitations and Future Work*

The IEQ in an office building may be influenced by many miscellaneous factors that are difficult to attribute (e.g., those arising as a function of mechanical operation, managerial strategy, and personal behaviour) [49], the direct influence experienced by the occupants of the building may result from a combination or interaction of several factors. Thus, directly correlating and comparing IEQ parameters to discomfort sources or symptoms may be slightly inaccurate. However, for this study, this has been done based on a literature review and the highest influencing parameters were accordingly selected. Moreover, this study was performed during the 3 months of winter 2019–2020 (December–February), although it was planned to be repeated in the summer months to have a more comprehensive collection of data. Due to the COVID-19 pandemic in 2020, the building was mostly unoccupied, and the plan was put off. Future work plans to study the summer conditions separately as it is a great concern in such a hot—arid climate, where international standards may not be suitable or agreed upon by the local building occupants. Additionally, it can further investigate the relationship between the variation of indoor environmental parameters and the subjective evaluations such as demographic, and detailed interviews with FM and building occupants.

**5. Conclusions**

This paper highlights the importance of the IEQ, especially within office buildings. As occupants spend almost 90% of their days indoors, the IEQ has many effects on the health and wellbeing of the occupants as well as their productivity level [6]. Moreover, poor IEQ conditions trigger an increased risk of sick building syndrome symptoms such as eye, nose, and throat irritations, allergies, headaches, fatigue, asthma-like symptoms, and several more [27]. The use of a comprehensive POE helps to examine and discover IEQ faults, allowing them to be tracked and modified to deliver positive effects on the occupant's satisfaction, on overall comfort, as well as to increase companies' productivity level and other relative economic profits.

As adopted from the Indoor Environmental Handbook (Bluyssen, 2009), the IEQ can be measured through four main factors namely; thermal comfort, indoor air quality (IAQ), lighting quality, and acoustical quality. A case study higher education office building in Al Ain, UAE, has been selected to collect field measurements and distribute relative questionnaires through the period from December to February (winter 2019/2020), to analyse and compare them in terms of occupants' satisfaction with the indoor environment, as well as to identify health-related symptoms experienced in the workspaces. The thermal comfort conditions revealed major concerns with the temperature level as 99% of the measurements were below an international standard (ASHRAE 55), and it was relatively emphasised with 55% of the survey respondents reporting "too cold" for the indoor thermal conditions. Contacting this information with the facility management team is advised to modify the set temperature in the building to a higher temperature within the comfort range. Similarly, providing the building user with a minor level of control to adjust the internal temperature will make for better thermal comfort conditions, achieving better user comfort, performance, and possible cooling energy reduction in this case. For the IAQ, PM2.5, PM10, $CO_2$, and TVOCs' levels have been monitored. Although the measured amounts show negligible concerns, 45% and 30% of the respondents reported "stuffy air" and "headache", respectively. This indicates that these symptoms could be tracked to other parameters or a combination of several. The measured lux levels regarding the lighting quality in the open offices reveal to be excessive with 100% above recommendations from the WELL building standard. However, only 19% of the occupants reported "too much artificial lighting". A lighting layout study is advised to be done to optimize the use of artificial lighting with daylighting and its further effects on occupant satisfaction as well as

potential energy reduction. Acoustic quality in the open office areas showed the highest respondents' dissatisfaction score (1.2 out of 3), which was expected due to the open office's layout along the corridors and lack of sound insulation. It needs to be mentioned how this paper's findings could be implemented in other buildings in the region. This case study building is one of the United Arab Emirates University (UAEU) buildings and the UAEU campus is managed by one FM (Facility Management) team named "Khadamat Facilities Management LLC", which means all other university buildings operate like the selected case study building. So, the common deficiencies found in this case study would have happened to other university buildings. Therefore, the findings from this paper could be applied to other university buildings to improve their IEQ for occupants, which could also be extended to other regions.

Keeping in mind the unique hot arid climate of the UAE, these results initiated a set of questions for further study of the IEQ during the summer months. The future of this research aims to identify recurrent issues in the IEQ and its relation to occupant satisfaction to enhance the facility management of the building to better serve the users as well as to find potential savings in the annual energy consumption. Moreover, issues relating to the layout of the building can help architects acknowledge the benefits and avoid repeating them by informing future designs.

**Author Contributions:** Conceptualization, Y.K.K.; methodology, Y.K.K.; software, Y.A.; validation, Y.K.K., A.A. and H.A.; formal analysis, Y.K.K. and Y.A.; investigation, Y.K.K.; resources, Y.K.K.; data curation, Y.A.; writing—original draft preparation, Y.A.; writing—review and editing, Y.K.K. and H.A.; visualization, Y.A. and A.A.; supervision, Y.K.K.; project administration, Y.K.K.; funding acquisition, Y.K.K. All authors have read and agreed to the published version of the manuscript.

**Funding:** This research was funded by the United Arab Emirates University (UAEU) grant number [G00002953] And The APC was funded by UAEU.

**Institutional Review Board Statement:** Not applicable.

**Informed Consent Statement:** Informed consent was obtained from all subjects involved in the study.

**Acknowledgments:** The authors would like to acknowledge the great support by the United Arab Emirates University (UAEU) for funding this study under the research grant STARTUP 2018 (G00002953) and also allowing the use of its university building for research.

**Conflicts of Interest:** The authors declare no conflict of interest.

## Appendix A. QUESTIONNAIRE FOR OCCUPANTS—Indoor Air Quality (IAQ)

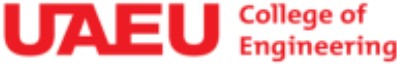

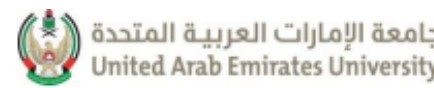

United Arab Emirates University      College of Engineering      Architectural Engineering

**QUESTIONNAIRE FOR OCCUPANTS – Indoor Air Quality (IAQ)**

Indoor Air Quality (IAQ) refers to the air quality within a building, especially as it relates to the health and comfort of building occupants. As humans spend a significant amount of time indoors; particularly in the workplace, the indoor air quality may affect their productivity level and well-being in the office. Thus, this research aims to understand, study, and find solutions to enhance the IAQ in office buildings.

**General information**

1. Age range:   ☐10-20   ☐20-30   ☐30-40   ☐40-50   ☐50-60

2. Gender: ☐ Male ☐ Female        3. Office no: . . . (floor/ room number)

4. What is the biggest part of the work you do? Please tick a box

    ☐ Managing people or resources

    ☐ Research work

    ☐ Using specialist skill (e.g. legal, engineering, scientific)

    ☐ Doing clerical, secretarial or administrative work

    ☐ Other, please write in ……………………………………………………………………………….

5. How long have you been working in this room?     . . . years   . . . months

6. How many days per week do you normally work at your desk     . . . days

7. How many hours per day do you normally work at your desk?     . . . hours

8. How many hours per day do you normally operate a PC at work?     . . . hours

**How often do you feel annoyed (uncomfortable) by the following? (tick one box)**

|  |  | often | regularly | sometimes | never |
|---|---|---|---|---|---|
| 9. | Dry air | ☐ | ☐ | ☐ | ☐ |
| 10. | Stuffy/ bad smell | ☐ | ☐ | ☐ | ☐ |
|  | From/where: | 1.outside/ 2. inside/ 3. Stairways and landings / 4. Toilets / 5. heating system/ 6. ventilation system/ 7. Other (circle possibilities) | | | |
| 11. | Static electricity | ☐ | ☐ | ☐ | ☐ |

| | | often | regularly | sometimes | never |
|---|---|:---:|:---:|:---:|:---:|
| 12. | Draught/ unpleasant cold breeze | ☐ | ☐ | ☐ | ☐ |
| | From/where: | 1.Windows/ 2. Stairways and landings / 3. Office door / 4. External wall / 5. ventilation system / 6. heating system/ 7. ceiling/ 8. other (circle possibilities) | | | |
| 13. | Too cold | ☐ | ☐ | ☐ | ☐ |
| | When: | 1.Winter/ 2. spring / 3. summer / 4. Autumn (circle possibilities) | | | |
| 14. | Too warm | ☐ | ☐ | ☐ | ☐ |
| | When: | 1.Winter/ 2. spring / 3. summer / 4. Autumn (circle possibilities) | | | |
| 15. | Temperature changes during a working day | ☐ | ☐ | ☐ | ☐ |
| 16. | Cold feet | ☐ | ☐ | ☐ | ☐ |

| | | often | regularly | sometimes | never |
|---|---|:---:|:---:|:---:|:---:|
| 17. | Warm surface | ☐ | ☐ | ☐ | ☐ |
| | Where: | 1. ceiling/ 2. Outer wall/ 3. windows / 4. Floor (warm feet)/ 5. Other (circle possibilities) | | | |
| 18. | Too much or too strong light | ☐ | ☐ | ☐ | ☐ |
| | Why: | 1. too much artificial light/ 2. Too much daylight/ 3. other (circle possibilities) | | | |
| 19. | Insufficient light | ☐ | ☐ | ☐ | ☐ |
| | Why: | 1. too little artificial light/ 2. Bad quality of lighting system/ 3. Too little daylight/ 4. other (circle possibilities) | | | |
| 20. | Reflections or glare | ☐ | ☐ | ☐ | ☐ |
| | Caused by: | 1. windows/ 2. Lighting system/ 3. other (circle possibilities) | | | |
| 21. | Unacceptable view | ☐ | ☐ | ☐ | ☐ |
| 22. | Feeling closed in | ☐ | ☐ | ☐ | ☐ |

|  |  | often | regularly | sometimes | never |
|---|---|:---:|:---:|:---:|:---:|
| 23. | Noise | ☐ | ☐ | ☐ | ☐ |

From: From outside/ 2. Adjacent rooms/ 3. Offices below/ 4. Offices above/ 5. Toilets/ 6. Stairways and corridors/ 7. Heating system/ 8. Ventilation system/ 9. Lifts/ 10. Escalators/ 11. Mail elevators/ 12. Automatic distribution system/ 13. Cleaning system/ 14. Colleagues in the office/ 15. Equipment in the office / 16. Machinery in building/ 17. Other (circle possibility)

**How much control do you feel you have over the following? (tick one box)**

|  |  | Not enough | Little | Reasonable | enough |
|---|---|:---:|:---:|:---:|:---:|
| 24. | Temperature | ☐ | ☐ | ☐ | ☐ |
| 25. | Ventilation | ☐ | ☐ | ☐ | ☐ |
| 26. | Light | ☐ | ☐ | ☐ | ☐ |

**If you are at the office for more than 4 hours, do you experience any of the following symptoms? (tick one box)**

|  |  | often | regularly | sometimes | never |
|---|---|:---:|:---:|:---:|:---:|
| 27. | Dry/watering eyes | ☐ | ☐ | ☐ | ☐ |
| 28. | Blocked/ runny nose | ☐ | ☐ | ☐ | ☐ |
| 29. | Dry/ irritated throat | ☐ | ☐ | ☐ | ☐ |
| 30. | Chest tightness | ☐ | ☐ | ☐ | ☐ |
| 31. | Dry/ irritated skin | ☐ | ☐ | ☐ | ☐ |
| 32. | Headaches | ☐ | ☐ | ☐ | ☐ |
| 33. | Lethargy/ tiredness | ☐ | ☐ | ☐ | ☐ |
| 34. | Pain n neck, shoulders, or back | ☐ | ☐ | ☐ | ☐ |

**About your room**

35. How many other people normally are in the room where you work? ⬜⬜ People

36. Which of the following equipment /items are present in your office room?

 PC/(laser)printer /humidifier/ionizer/plants/other **(circle possibilities)**

**About yourself**

37. Have you ever suffered from fever or other allergic reactions? ⬜ yes ⬜ no

38. Have you ever had asthmatic problems? ⬜ yes ⬜ no

39. Have you ever suffered from eczema? ⬜ yes ⬜ no

40. Do you mind us visiting your office? ⬜ yes ⬜ no

41. If you have any comments or remarks you can put them here.

----------------------------------------------------------------------------------------------------

----------------------------------------------------------------------------------------------------

----------------------------------------------------------------------------------------------------

----------------------------------------------------------------------------------------------------

----------------------------------------------------------------------------------------------------

----------------------------------------------------------------------------------------------------

----------------------------------------------------------------------------------------------------

**Thank you for your time!!**

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
