# Peer review of "Indoor Environmental Quality Assessment and Occupant Satisfaction: A Post-Occupancy Evaluation of a UAE University Office Building"

_buildings, doi:10.3390/buildings12070986_

Round 1
Reviewer 1 Report
This article deals with evaluation of the quality of the environment of an university office by detailed measurement of the indoor environment and a questionnaire by the occupants. It is described carefully. But the important point, which is missing, is the environmental target at the design stage. The building performance and equipment system was supposed to be designed by setting what kind of environment should be realized. For example, it was assumed that ventilation airflow rate was decided. Therefore, authors should measure the ventilation rate and check it. It is necessary to clarify firstly whether the realized environment meets the design targets. In addition, it is necessary to clearly indicate the weather conditions of the outside environment such as the outside air temperature, relative humidity, and the amount of solar radiation during the measured period.
Reviewer 2 Report
11. It is understood that due to the COVID-19 pandemic in 2020, the building in this study was mostly unoccupied, and the plan for summer measurement was put off. As a result, the title of this paper should be revised (Indoor Environmental Quality Assessment and Occupant Satisfaction: A Post Occupancy Evaluation of a Higher Education Office Building in Hot Arid Climate). In hot Arid climate? Or in Arid winter climate?
2. The authors claimed that “The main outcome of this research contributed to identifying correlations between measured data and occupant satisfaction, and at the same time, identifying common IEQ defects in order to better communicate with facility managers and architects”. It seems that the work reported in this paper is a case study. The authors should discuss how the results can be applied in other buildings or regions.
Reviewer 3 Report
The paper “Indoor Environmental Quality Assessment and Occupant Satisfaction: A Post Occupancy Evaluation of a Higher Education Office Building in Hot Arid Climate” is very interesting. This paper is likely to contribute to the field of study related to the indoor environment. However, I still have several comments or questions need to be addressed before this review could be accepted for publication. Detailed comments and questions are given in the section that follows.
Major comments:
- Section 2.3. User satisfaction questionnaire. “The perception of the respondents was measured using a standardized 4-point scale of frequency. As it was essential to find specific user opinions in this survey, the 4-point scale was found to be most ideal.” Was a Likert-scale used? Please specify in this section the questions contained in the survey questionnaire.
- Frequency score of sources contributing to discomfort is reported in the Result section. However, there is no mention of how this data has been obtained in the Methodology section. What question has been included in the questionnaire for this purpose? Is it an open-ended question or a multiple choice question? This issue needs to be clarified in the methodology section.
- Result section. Page 12. “Respondents were asked how often they experience a set of sources contributing to uncomfortable sensations. as mentioned earlier, the questions followed a standardized format consisting of a 4-point scale from which the respondent can choose a rank equivalent to their experienced sensation. The options provided were: never (neutral), sometimes (slightly uncomfortable), regularly (moderately uncomfortable), and often (extremely un-comfortable). For analysis, the rank selected from the 4-point scale in each question was transformed into an integer between 0 and 3.” This paragraph is not results data and the information should be moved to the Methodology section.
- Page 17. “These results highlight the correlation between the average temperature value measured (21.5°C) and the occupants perceived sensation (“too cold”), as well as validate the recommended range standard set by ASHRAE-55 for the winter in hot arid climate (24°C - 26°C).” How has this correlation analysis been carried out? Detailed information on this analysis should be added to the manuscript. I suggest the authors to clearly illustrate in the text.
- The results show us the detailed assessments on indoor environment. I think the authors can further investigate the relationship between the variation of indoor environmental parameters and the subjective evaluations. It would be better to add a “Statistical Analysis” section as most papers have.
Minor comments:
- Abstract – The current “Results” section in “Abstract” is over-simple, without any # or values shown and without any results described. Please strengthen this part.
- Section 2.1. Case study building. Sentence: “…extremely hot summers (38 C average) and warm winters (18 C average)…” Please indicate the correct temperature unit.
- Information is missing in Figure 2. There is no identification of the measurement points (FF1, FF4, etc.) Since in Section 3 the results are expressed with the measurement points, please modify Figure 2 and add the location of these points.
- In section 3.1, the style subsubsection of the template should be used; i.e. “3.1.1. Thermal comfort.”
- Regarding Figure 3, the figure legend is missing. Currently, it is not possible to identify the data obtained from each sensor with the values of the graph.
- The resolution of Figure 4 is low. Please improve it.
- Section 4. Discussion. The style subsection of the template should be used; i.e. “4.1 Thermal comfort.”
Round 2
Reviewer 1 Report
Authors replied to reviewer's comments properly.
Author Response
Thank you very much for your comments and recommendation!
Reviewer 3 Report
The manuscript is better after the review process. However, the Abstract exceeds the words limit (> 200 words).
Author Response
Rewrite the abstract and reduced it from 266 words to 213 words.
Hope this would be OK for now and thank you so much for your great comments and suggestions.
Please, find the rewritten abstract below
Abstract: As occupants spend almost 90% of their day indoors, especially in the workplaces, Indoor Environmental Quality (IEQ) plays a primary role in health and wellbeing, productivity, and building energy consumption. Adopting the IEQ and Post Occupancy Evaluation (POE), data has been gathered from nine multilevel open offices within a university building located in Al Ain, in the United Arab Emirates (UAE) for three winter months. Physical parameters were monitored using data loggers to record the main IEQ factors. In parallel, POE questionnaires have been distributed to obtain occupants’ satisfaction with the IEQ and health-related symptoms experienced in the workspaces. The IEQ and POE data have shown slightly above or below the recommended ranges with the occupants similarly and slightly dissatisfied with the building. The thermal comfort revealed concerns with 99% of temperatures below international standards where 55% of the survey respondents reported “too cold”. The IAQ measurements showed 45% and 30% of the respondents reporting “stuffy air” and “headache” which indicated symptoms that could be tracked to other parameters or a combination of several, and the findings have been discussed in detail in this paper. This research contributed to identifying correlations between measured data and occupant satisfaction and identifying common IEQ defects and their sources to better communicate with facility managers and architects.